# OCIM: Object-centric Compositional Imagination for Visual Abstract Reasoning

## Abstract

A long-sought property of machine learning systems is the ability to compose learned concepts in novel ways that would enable them to make sense of new situations. Such capacity for imagination – a core aspect of human intelligence – is not yet attained for machines. In this work, we show that object-centric inductive biases can be leveraged to derive an imagination-based learning framework that achieves compositional generalization on a series of tasks. Our method, denoted **O**bject-centric **C**ompositional **Im**agination (OCIM), decomposes visual reasoning tasks into a series of primitives applied to objects without using a domain-specific language. We show that these primitives can be recomposed to generate new imaginary tasks. By training on such imagined tasks, the model learns to reuse the previously-learned concepts to systematically generalize at test time. We test our model on a series of arithmetic tasks where the model has to infer the sequence of operations (programs) applied to a series of inputs. We find that *imagination is key* for the model to find the correct solution for unseen combinations of operations.

## 1 Introduction

Humans have the remarkable ability to adapt to new unseen environments with little experience (Lake et al., 2017). In contrast, machine learning systems are sensitive to distribution shifts (Arjovsky et al., 2019; Su et al., 2019; Engstrom et al., 2019). One of the key aspects that makes human learning so robust is the ability to produce or acquire new knowledge by composing few learned concepts in novel ways, an ability known as compositional generalization (Fodor and Pylyshyn, 1988; Lake et al., 2017). Although the question of how to achieve such compositional generalization in brains or machines is an active area of research (Ruis and Lake, 2022), a promising hypothesis is that dreams are a crucial element (Hoel, 2021) through the Overfitted Brain Hypothesis (OBH).

Both imagination and abstraction are core to human intelligence. Objects in particular are an important representation used by the human brain when applying analogical reasoning (Spelke, 2000). For instance, we can infer the properties of a new object by transferring our knowledge of these properties from similar objects (Mitchell, 2021). This realization has inspired a recent body of work that focuses on learning models that discover objects in a visual scene without supervision (Eslami et al., 2016b; Kosiorek et al., 2018; Greff et al., 2017; van Steenkiste et al., 2018; Greff et al., 2019; Burgess et al., 2019; van Steenkiste et al., 2019; Locatello et al., 2020). Many of these works propose several inductive biases that lead to a visual scene decomposition in terms of its constituting objects. The expectation is that such an object-centric decomposition would lead to better generalization since it better represents the underlying structure of the physical world (Parascandolo et al., 2018). To the best of our knowledge, the effect of object-centric representations for systematic generalization in visual reasoning tasks remains largely unexplored.

While abstractions, like objects, allow for reasoning and planning beyond direct experience, novel configurations of experienced concepts are possible through imagination. Hoel (2021) goes even further and posits that dreaming, which is a form of imagination, improves the generalization and robustness of learned representations. Dreams do so by producing new perceptual events that are composed of concepts experienced/learned during wake-time. These perceptual events can be described by two knowledge types (Goyal et al., 2020; 2021b): the declarative knowledge encoding object states (e.g. entities that constitute the dreams), and the procedural knowledge encoding how they behave and interact with each other (e.g. how these entities are processed to form the percep-

tual event). In this work, we take a step towards showing how OBH can be implemented to derive a new imagination-based learning framework that allows for better compositional generalization like dreams do.

We thus propose OCIM, an example of how object-centric inductive biases can be exploited to derive imagination-based learning frameworks. More specifically, we model a perceptual event by its object-centric representations and a modular architecture that processes them to solve the task at hand. Similar to (Ellis et al., 2021), we take the program-induction approach to reasoning. In order to solve a task, the model needs to (1) abstract the perceptual input in an object-centric manner (e.g. represent declarative knowledge), and (2) select the right arrangement of processing modules (which can be seen as a *neural program*) that solves the task at hand. In order to generalize beyond direct experience through imagined scenarios, a model would have to imagine both of these components (e.g. objects + how to process them). Here we restrict ourselves to imagining new ways to process experienced perceptual objects. We propose to do so by exploiting object-centric processing of inductive biases. The idea is to have a *neural program* (Reed and De Freitas, 2015; Cai et al., 2017; Li et al., 2020) composed of modular neural components that can be rearranged (e.g. "sampled" through selection bottlenecks) to invent new tasks. The capacity to generate unseen tasks enables OCIM to generalize systematically to never-seen-before tasks by (1) producing new imagined scenarios composed of learned/experienced concepts and (2) training the model on these imagined samples to predict back their constituting concepts (e.g. used modules that were sampled to produce them). Our contribution is threefold:

- We propose an example of how object-centric inductive biases can be used to derive an *imagination*-based learning framework. Specifically we show that rearranging modular parts of an object-centric processing model to produce an imagined sample and training the model to predict the arrangement that produced that sample helps with compositional generalization.
- We propose a visual abstract reasoning dataset to illustrate our imagination framework and evaluate the models along different axis of generalization.
- We highlight some drawbacks of current state-of-the-art (SOTA) object-centric perception model when it comes to disentangling independent factors of variation within a single visual object.

## 2 RELATED WORK

**Object-centric Representation.** A recent research direction explores unsupervised object-centric representation learning from visual inputs (Locatello et al., 2020; Burgess et al., 2019; Greff et al., 2019; Eslami et al., 2016a; Crawford and Pineau, 2019; Stelzner et al., 2019; Lin et al., 2020; Geirhos et al., 2019). The main motivation behind this line of work is to disentangle a latent representation in terms of objects composing the visual scene (e.g. slots). Recent approaches to slot-based representation learning focus on the generative abilities of the models; in our case, we study the impact of object-centric inductive biases on systematic generalization of the models in a visual reasoning task. We observe that modularity of representations is as important as the mechanisms that operate on them (Goyal et al., 2020; 2021b). Additionally, we show that object-centric inductive biases of both representations and mechanisms allow us to derive an imagination framework that leads to better systematic generalization.

**Modularity.** Extensive work from the cognitive neuroscience literature (Baars, 1997; Dehaene et al., 2017) suggests that the human brain represents knowledge in a modular way, with different parts (e.g, modules) interacting with a working memory bottleneck via attention mechanisms. Following these observations, a line of work in machine learning Goyal and Bengio (2020); Goyal et al. (2019; 2020; 2021b); Ostapenko et al. (2021); Goyal and Bengio (2022) has proposed to translate these characteristics into architectural inductive biases for deep neural networks. Recent approaches have explored architectures composed of a set of independently parameterized modules that compete with each other to communicate and attend or process an input (Goyal et al., 2019; 2020; 2021b). Such architectures are inspired by the notion of independent mechanisms (Pearl, 2009; Bengio et al., 2019; Goyal et al., 2019; Goyal and Bengio, 2022), which suggests that a set of independently parameterized modules capturing causal mechanisms should remain robust to distribution

shifts caused by interventions, as adapting one module should not require adapting the others. The hope is that out-of-distribution (OOD) generalization would be facilitated by making it possible to sequentially compose the computations performed by these modules, whereby new situations can be explained by novel combinations of existing concepts. In this work, we show how modular architectural choices can be exploited to derive an imagination-based learned paradigm that allows better compositional generalization; we do so by explicitly exposing the model to data samples composed of novel combination of learned concepts.

**Imagination, Dreaming, and Generalization** Dreams are a form of imagination that have inspired a significant amount of influential work (Hinton et al., 2006; Ellis et al., 2021; Hafner et al., 2019; 2020). An interesting explanation for such phenomenon is the *overfitted brain hypothesis* (OBH) (Hoel, 2021), which states that dreaming improves the generalization and robustness of learned representations. The idea is that, while dreaming, the brain recombines patterns seen during wake time. This results in artificial data augmentation in the form of dreams. This way, dreams regularize and prevent the brain from overfitting the patterns seen while being awake. In machine learning, data augmentation is a long standing technique for tackling the data scarcity problem, whereby new training samples are generated from existing data in order to diversify the trained models. Various approaches for augmentation, such as GANs (dos Santos Tanaka and Aranha, 2019), VAEs (Chadebec and Allassonnière, 2021), and diffusion models Ho et al. (2022) have proved to be effective in improving model accuracy and generalizability. As in the case of dreams for the human brain, the strategy of considering imagined data samples is broadly compatible with data augmentation objectives. Dreamcoder (Ellis et al., 2021) is a recent example where training on imagined patterns improve generalization for program induction. Program induction is a challenging problem because the search space is combinatorially large and new unseen programs have low likelihood. To address these challenges, Dreamcoder leverages a wake-sleep algorithm that reduces the search space by learning a domain-specific language (DSL) while learning to search programs efficiently. During training, Dreamcoder undergoes a *dreaming* phase where the model learns to solve new tasks generated by sampling programs from a DSL and applying them to inputs seen during the *wake* phase. Although Dreamcoder is promising for program induction, the DSL is a major roadblock to solve open-ended visual reasoning tasks where the input consists of raw pixels rather than symbols. In this work, we overcome these challenges by relying on object-centric inductive biases, i.e., architectural choices that both represent and process objects, in order to *learn* a neural program library.

## 3 VISUAL ARITHMETIC REASONING DATASET

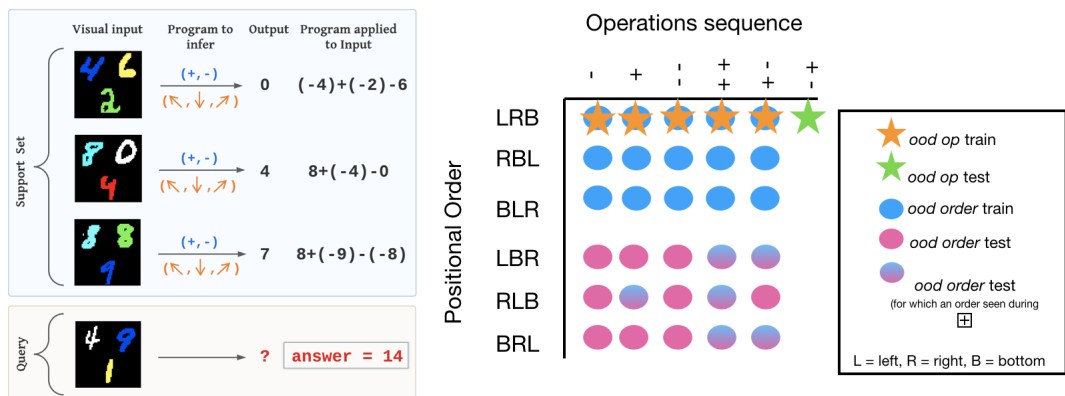

Figure 1: Data sample and dataset splits.

Most visual reasoning benchmarks revolve around variations of Raven's Progressive Matrices (RPM) (James, 1936; Zhang et al., 2019; Barrett et al., 2018; Hoshen and Werman, 2017) which are *discriminative* tasks in which the solver chooses from a set of candidate answers. However, in a recent survey, Mitchell (2021) recommends evaluating models on generative tasks that focus on

human core knowledge (Spelke, 2000). Models trained on generative tasks are indeeed less prone to shortcut learning and systems that generate answers are in many cases more interpretable. To that end Chollet (2019) proposes the Abstract Reasoning Corpus (ARC), where the model is given a few examples of Input-Output (I/O) pairs and has to understand the underlying common program that was applied to the inputs to obtain the outputs. ARC tasks are meant to rely only on the innate core knowledge systems which include intuitive knowledge about objects, agents and their goals, numerosity, and basic spatial-temporal concepts. However, ARC remains unapproachable by current deep learning methods.

We propose to take a step towards solving ARC by designing a new generative benchmark in which we evaluate systematic compositional generalization. Like ARC, our dataset is composed of a collection of support sets, each having a number of input/output pairs, such that the output for every sample (support set) is obtained by applying the same *program* to the corresponding input. The model is then presented with a new query input and evaluated on its ability to predict the right associated output (i.e. applying the inferred program in the support set to the query input). The inputs are $56 \times 56$ images with three colored MNIST digits placed at three different positions. These visual digits can have values between $-9$ and $9$ and their color represents their sign. There are six different colors in total (3 of them are *negative* and the remaining 3 are *positive*). The program applied to the inputs is a sequence of arithmetic operations (we restrict ourselves to addition and subtraction and the dataset can further be extended with more complicated queries that involve comparison between the different objects, maximum operations etc..) in a particular positional order. Since we are interested in the model's ability to generalize compositionally to unseen examples, we create different splits that aim at evaluating different axes of compositional generalization. These three splits are as follows : (1) OOD seq : In this split, during training we leave out some sequence of operations (e.g. $(+, -)$) and perform the evaluation on samples requiring the excluded sequence., (2) OOD order: where uring training the model only sees programs that take input digits in some particular positional order (e.g. top-left, top-right, down) and is evaluated on unseen orders., and (3) OOD perception which evaluate the perception module ability to disentangle the digit class (e.g. 1 to 9) from its color (representing its sign). We thus consider certain pairs of digit-color configurations during training and evaluate the model on unseen pairs.

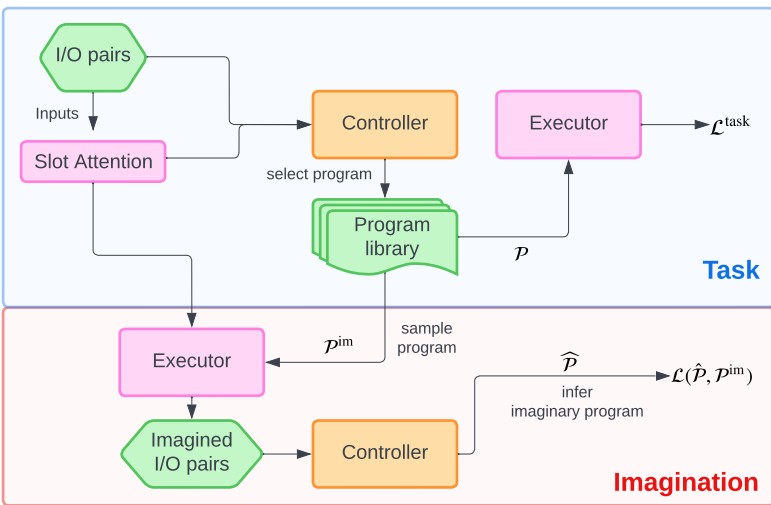

Figure 2: Task and Imagination pathways in OCIM

## 4 OBJECT-CENTRIC COMPOSITIONAL IMAGINATION (OCIM)

Our model is designed to generate answers in a sequential way. The design choices that we make reflect the fact that the output answer to a generative reasoning task can be computed by sequentially updating a working memory whose arguments are obtained from available input (e.g. slots extracted from images in our case). The computation steps are the following: (1) Visual inputs **x** are mapped

to $N_s$ object-centric slots $\mathbf{S} = [\mathbf{S}_1, .., \mathbf{S}_{N_s}]$ using a Slot Attention module (Locatello et al., 2020), then, (2) the controller takes the support set as input and ouputs a single task embedding $\mathbf{z}$; this task embedding is then (3) translated into a sequential neural program (e.g. a sequence of transformation to be applied to a working memory); finally, (4) the executor takes this neural program along with an input query (i.e. its object-centric slots) and performs the sequential updates. The overall computation paths are given in Figure 2.

The main contribution of our work resides in the architecture of the executor, its interface with the controller (i.e. how the executor uses the information encoded in the controller) and the derived imagination machinery. The controller scaffold (detailed in the appendix) that we use in all baselines always outputs a single task embedding $\mathbf{z}$ and can be adapted depending on the task at hand. In this section, we detail the modeling choices of the executor and the interface between the executor and the controller, which is formulated as a *selection bottleneck* and the imagination component.

## 4.1 EXECUTOR

The executor takes a visual query input $\mathbf{x} \in \mathbb{R}^{56 \times 56 \times 3}$ and a *neural program*; it then updates a working memory $\mathbf{h} \in \mathbb{R}^p$ in a sequential and structured manner. The visual input is first mapped to a set of $N_s$ object-centric slots $[\mathbf{S}_1, \mathbf{S}_{N_s}]$ that is later used as candidate arguments for each update of the working memory. The executor is composed of a library of $N_r$ learned modules (e.g. rules, implemented as small GRU cells) and $N_c$ condition values. The conditions are expected to encode the way in which to select an argument (e.g. among the slots) that is in turn used by a module to update the working memory. Both rules (i.e. modules) and conditions are indexed by some learned tags $\mathbf{M} = [\mathbf{M}_1, \ldots, \mathbf{M}_{N_r}]$ and $\mathbf{C} = [\mathbf{C}_1, \ldots, \mathbf{C}_{N_c}]$. The *neural program* that the executor takes as input generates (1) the number $T$ of updates that the executor needs to perform, specified by a scalar gate at each time step; we denote the sequence of such gates by $g = [g_1, \ldots, g_T])$ ; (2) the sequence of modules $[\hat{m}_1, \ldots, \hat{m}_T]$ (each $\hat{m}_t$ parameterized by a small learned GRU RNN) that will perform the $T$ updates of the working memory; and (3) the sequence of conditions $[\hat{c}_0, \ldots, \hat{c}_T]$ (each $\hat{c}_t$ being a condition vector that selects one slot; this slot will be used as an argument to the selected module at each time step). At each time step $t$, each update in the sequence is done in the following two steps:

- **Argument selection**: given a condition vector $\hat{c}_t$ select an argument of the update from among the input slots $\mathbf{S}$ of the query.
- **Update**: update working memory $h_{t-1}$ with GRU rule $\hat{m}_t$ and the selected argument $\hat{s}_t$.

**Argument Selection.** At each time step, a slot argument is selected through a key-query attention mechanism. The idea is that the condition vector $\hat{c}_t$ is compared against all the input slots to select the one that corresponds best to the features encoded in the condition (e.g. select the slot at the "top-left" of the image). The attention mechanism is thus realized using the condition vector $\hat{c}_t \in \mathbb{R}^{1 \times d}$ as a query and the $N_s$ slots $\mathbf{S} = [\mathbf{S}_1, \ldots, \mathbf{S}_{N_s}] \in \mathbb{R}^{N_s \times d}$ as keys such that the selected argument $\hat{s}_t$ at time-step $t$ is given by:

$$\hat{s}_t = \text{GumbelSoftmax}(\frac{\hat{c}_t \mathbf{S}^T}{\sqrt{d}})\mathbf{S} \in \mathbb{R}^{1 \times d} \tag{1}$$

The sequence of selected arguments is thus given by $\hat{\mathbf{s}} = [\hat{\mathbf{s}}_0, \ldots, \hat{\mathbf{s}}_T]$.

**Sequential Update.** Given a sequence of processing modules $[\hat{m}_1, \ldots, \hat{m}_T]$, a sequence of input arguments $[\hat{\mathbf{s}}_0, \ldots, \hat{s}_T]$ and a length given by a sequence of gates $[g_1, .., g_T]$, the executor updates a working memory whose state at time step $t$ is denoted by $\mathbf{h}_t$ such that:

$$\mathbf{h}_{t+1} = (g_{t+1})\mathbf{h}_t + (1 - g_{t+1})\hat{\mathbf{m}}_t(\hat{\mathbf{s}}_{t+1}, \mathbf{h}_t) \text{ and } \mathbf{h}_0 = \hat{\mathbf{s}}_0 \tag{2}$$

For ease of notation, we let $\textbf{Executor}(\mathbf{x}, \mathcal{P})$ be the result of applying the neural program $\mathcal{P}$ to the visual input $\mathbf{x}$.

## 4.2 SELECTION BOTTLENECK

In this section, we describe the interface between the controller and the executor: how the task embedding $\mathbf{z}$ output by the controller is transformed into a *neural program* that the executor then

takes as input (e.g. sequences of modules, conditions and gates) to perform the sequential update. First the task embedding $\mathbf{z} \in \mathbb{R}^d$ is transformed into a sequence of embeddings by feeding $\mathbf{z}$ as argument to a GRU RNN that starts with an empty hidden state $[\mathbf{z}_1, \ldots, \mathbf{z}_T] = \text{GRU}(\mathbf{z})$.

Both module and condition selections are done through a key-query attention mechanism comparing the task embedding $\mathbf{z}_t$ to the $N_r$ learned module tags (denoted by $\mathbf{M} = [\mathbf{M}_1, \ldots, \mathbf{M}_{N_r}] \in \mathbb{R}^{N_r \times d}$) and the $N_c$ learned conditions tags (denoted by $\mathbf{C} = [\mathbf{C}_1, \ldots, \mathbf{C}_{N_c}] \in \mathbb{R}^{N_c \times d}$). The keys are extracted from the condition tags, whereas the query is extracted in both attention operations form the task embedding $\mathbf{z}_t$ (using two MLPs $Q_r$ and $Q_c$) such that the $t$-th element of each sequence is obtained with:

$$W_m^t = \text{GumbelSoftmax}\left(\frac{Q_r(\mathbf{z_t})\mathbf{M}^T}{\sqrt{d}}\right) \in \mathbb{R}^{1 \times N_r} \tag{3}$$

and the resulting update is given by the following weighted sum $\hat{m}_t(\mathbf{h}_t, \hat{\mathbf{s}}_t) = \sum_{i=1}^{N_r} W_m^t[i] m_i(\mathbf{h}_t, \hat{\mathbf{s}}_t)$. Similarly, the conditions are obtained through:

$$\hat{\mathbf{c}}_t = \text{GumbelSoftmax}\left(\frac{Q_c(\mathbf{z_t})\mathbf{C}^T}{\sqrt{d}}\right)\mathbf{c} \tag{4}$$

with $\mathbf{c} \in \mathbb{R}^{N_c \times d}$ denoting the set of learned condition vectors.

Finally, the sequence of step gates are obtained directly from the sequence of $[\mathbf{z}_1, \ldots, \mathbf{z}_T]$ such that

$$g_t = \text{MLP}(\mathbf{z}_t) \tag{5}$$

For ease of notation, we let $\mathcal{P}_\mathbf{z} = \textbf{SelectionBottleneck}(\mathbf{z}) = \{\mathbf{g}, \hat{\mathbf{c}}, \hat{\mathbf{m}}\}$ denote the neural program obtained from the task embedding $\mathbf{z}$, where $\mathbf{g}, \hat{\mathbf{c}}, \hat{\mathbf{m}}$ correspond to the associated step gates, condition vectors and processing module sequences.

## 4.3 Compositional Imagination

Our main contribution resides in showing how object-centric inductive biases (used in the executor) can be leveraged to induce a new imagination-based learning framework that leads to better compositional generalization. The idea is that the same way we select a sequence of modules, conditions and gates using the task embedding output by the controller, we can also sample them at random (from a uniform distribution) to create a new neural program that can be used to create imagined scenarios. To do so, we sample at random a sequence of gates $\mathbf{g}^{\text{im}} = [g_1^{\text{im}}, \ldots, g_T^{\text{im}}]$, a sequence of condition vectors $\mathbf{c}^{\text{im}} = [\mathbf{c}_0^{\text{im}}, \ldots, \mathbf{c}_T^{\text{im}}]$ and a sequence of processing modules $\mathbf{m}^{\text{im}} = [\mathbf{m}_0^{\text{im}}, \ldots, \mathbf{m}_T^{\text{im}}]$ that correspond to the procedural part of the knowledge we have about the reasoning task at hand. Ideally we would also sample the query to process (e.g. the declarative part) but we leave that for future work. Instead, we take visual inputs that are already present in the training data and we apply an imagined neural program to them. Since the goal is to create new samples, we need to apply the same imagined program to a set of visual inputs to form a support set.

Let $\mathbf{X}^{supp} = \{\mathbf{x}_1, \ldots, \mathbf{x}_L\}$ denote a set of visual inputs from the training data, and let $\mathcal{P}^{\text{im}} = \{\mathbf{g}^{\text{im}}, \mathbf{c}^{\text{im}}, \mathbf{m}^{\text{im}}\}$ be an imagined program. Then the imagination phase can be split into 3 main steps:

- **Imagined samples**: this step consists of applying an imagined program $\mathcal{P}^{\text{im}}$ to a support set of visual input $\mathbf{X}^{supp}$ to obtain an imagined I/O support set $\mathcal{S}^{\text{im}} = \{\mathbf{X}^{supp}, \mathbf{O}^{im}\}$ with $\mathbf{O}^{im} = [\textbf{Executor}(\mathbf{x}_i, \mathcal{P}^{\text{im}})$ for $\mathbf{x}_i \in \mathbf{X}^{supp}]$.

- **Task embedding inference**: this step consists of encoding the imagined support set with the controller to produce a task embedding $\mathbf{z}^{\text{im}} = \textbf{Controller}(\mathcal{S}^{\text{im}})$.

- **Mechanisms Prediction**: the last step consists of predicting back the neural programs (i.e. its components) that produced the imagined sample. This means matching $\hat{\mathcal{P}}_{\mathbf{z}^{\text{im}}} = \textbf{SelectionBotlleneck}(\mathbf{z}^{\text{im}})$ with $\mathcal{P}^{\text{im}}$.

The associated loss is called the imagination loss $\mathcal{L}^{\text{im}} = \mathcal{L}(\hat{\mathcal{P}}_{\mathbf{z}^{\text{im}}}, \mathcal{P}^{\text{im}})$, which can be split into 3 cross-entropies predicting the step gate values, the conditions vector indices and the processing module indices. During training, we introduce this loss after a warming period during which the model is trained only on the training data available. We detail the hyperparameters in the Appendix.

---

**Algorithm 1** Executor Pseudocode

---

1: **function** EXECUTOR($\mathbf{x}, \mathcal{P}$)                    ▷ The input $\mathbf{x}$ and program $\mathcal{P}$
2:     $\mathbf{S} = \text{SlotAttention}(\mathbf{x})$                    ▷ Object-centric perception
3:     $\mathbf{g}_{1..T}, \hat{\mathbf{m}}_{1..T}, \hat{\mathbf{c}}_{0..T} \leftarrow \mathcal{P}$
4:     $\mathbf{h}_0 \leftarrow \text{GumbelSoftmax}(\frac{\hat{c}_0 \mathbf{S}^T}{\sqrt{d}})\mathbf{S}$          ▷ Working memory initialization
5:     **for** $(t = 1; t < T; t + +)$ **do**
6:         $\hat{\mathbf{s}}_t \leftarrow \text{GumbelSoftmax}(\frac{\hat{c}_t \mathbf{S}^T}{\sqrt{d}})\mathbf{S} \in \mathbb{R}^{1 \times d}$                    ▷ Eq. 1
7:         $\mathbf{h}_{t+1} \leftarrow (g_{t+1})\mathbf{h}_t + (1 - g_{t+1})\hat{\mathbf{m}}_t(\hat{\mathbf{s}}_{t+1}, \mathbf{h}_t)$          ▷ Eq. 2
8:     **end for**
9:     **return** $\text{pred}(\mathbf{h}_T)$                    ▷ Task-specific prediction
10: **end function**

---

**Algorithm 2** Compositional Imagination

---

**Require:** $\mathbf{X}^{supp}$                    ▷ Samples seen during training
1: $\mathbf{S}^{supp} \leftarrow \text{SlotAttention}(\mathbf{X}^{supp})$          ▷ Object-centric perception
2: $\mathcal{P}^{\text{im}} \sim U(\mathbf{g}, \mathbf{C}, \mathbf{M})$                    ▷ Sample a program
3: $\mathbf{O}^{im} = \{\text{Executor}(\mathbf{x}_i, \mathcal{P}^{\text{im}}) \text{ for } \mathbf{x}_i \in \mathbf{X}^{supp}\}$
4: $\mathcal{S}^{\text{im}} \leftarrow \{\mathbf{X}^{supp}, \mathbf{O}^{im}\}$
5: $\mathbf{z}^{\text{im}} \leftarrow \text{Controller}(\mathcal{S}^{\text{im}})$
6: $\widehat{\mathcal{P}}_{\mathbf{z}^{\text{im}}} = \text{SelectionBottleneck}(\mathbf{z}^{\text{im}})$                    ▷ Infer program
7: $loss_{\text{im}} \leftarrow \text{CrossEntropy}(\mathcal{P}, \widehat{\mathcal{P}}_{\mathbf{z}^{\text{im}}})$

---

## 4.4 TRAINING

The training of the whole model can be split into three phases: **Step 1**: Pretraining of the perception model such that the next steps start with reasonable latent slots. - **Step 2**: Regular training on the task prediction objective (8-binary-bits digit prediction). - **Step 3**: Imagination, where random modules and conditions are sampled to create new data points and expose the model to potentially OOD samples.

Each of these steps gives rise to a specific objective loss to optimize. The **task prediction** objective in our case is a simple cross-entropy on the output of the executor, since we treat each bit of the output as a binary label to predict. This loss is given by:

$$\mathcal{L}_{\text{task}} = -\sum_{i \in \mathcal{D}_{\text{train}}} (\mathbf{y}_i \log(\hat{\mathbf{y}}_i) + (1 - \mathbf{y}_i) \log(1 - \hat{\mathbf{y}}_i)). \tag{6}$$

The pretraining phase consists of training the Slot Attention module on a reconstruction task. During the imagination phase, new samples are created according to Algorithm 2, and the model is optimized to infer the programs that generated these samples and to minimize $\mathcal{L}^{\text{task}}$ at the same time. We detail the hyperparameters associated to the different training phases in the Appendix.

## 5 EXPERIMENTS

OCIM has two main components. The perception component and the object processing (i.e. reasoning) component. Our contribution lies in the object processing component, while for the perception component we use a SOTA slot attention module (Locatello et al., 2020). The goal of this section is two-fold: (1) to evaluate our imagination-based learning paradigm on a set of compositional generalization axes and (2) the ability of the perception module to extract symbolic-like representations that can be used to solve our visual abstract reasoning task.

## 5.1 BASELINES

Our model OCIM can be seen as an extension of the sparse interaction inductive biases proposed in Neural Production Systems (NPS) Goyal et al. (2021a), and augmented with an imagination-based

learning mechanism. NPS sequentially updates a set of slots by choosing at each time-step a primary slot, a secondary slot, and an interaction rule with some key-query attention mechanisms. OCIM sequentially updates the state of a shared working memory across time steps (e.g. instead of slot states) from which the final answer can be extracted. As a result, at each time-step the primary argument of an *interaction* is always the shared memory and the second argument is selected among the input slots. We also compare OCIM (and its variant without imagination that we call OCIM-noim) to two other baselines in which the executor is parameterized with a single monolithic GRU RNN in one case, and with a dense GNN in the other, for which we use the interaction component from the C-SWM (Kipf et al., 2019) model (like Goyal et al. (2021b)). In each of these baselines, input nodes correspond to extracted slots concatenated with the output of the controller. For the GNN baseline, a GRU RNN is added after computing the interactions between nodes to aggregate the final result. We refer to these two baselines as **GRU** RNN and **GNN** respectively, and detail their exact parameterization in the Appendix.

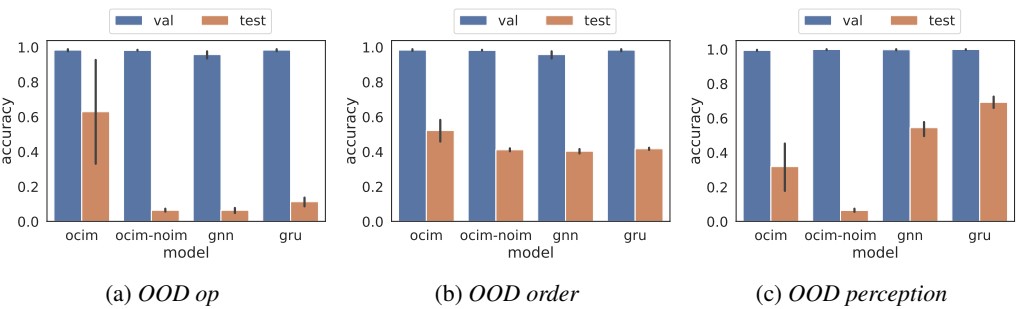

(a) *OOD op*       (b) *OOD order*       (c) *OOD perception*

Figure 3: Validation and test accuracy for the baselines and OCIM on the three axes of compositional generalization described in Section 3. Insights about the high variance of the results for OCIM is given in 5.3 when inspecting modules specialization.

## 5.2 OOD Splits

We are interested in 3 axes of compositional generalization that evaluate both our proposed imagination-based learning paradigm and the perception module robustness: (1) We first want to evaluate whether the imagination phase in OCIM can lead to a better generalization to arithmetic tasks composed of never-seen sequences of operations during training; (2) we then want to evaluate whether OCIM is able to generalize to never-seen orders in which the input digits are taken to perform the sequence of operations (e.g. its ability to extract meaningful and general argument selection conditions), and finally, (3) we want to evaluate whether current object-centric iductive biases as proposed in Slot Attention Locatello et al. (2020) are well suited for disentangling independent factors of variations within an object (e.g. color and digit class in our case). To that end, we propose to evaluate the models on the three splits described in Section 3:

## 5.3 Results

**Imagination and Generalization.** In our experiments, all the models share the same perception model Locatello et al. (2020) and the same controller. They only differ by their execution component. In Figure 3, we report the accuracy peformances of our model compared to the baselines of interest across three different splits that aim at evaluating a particular axis of generalization. We observe two main results: (1) Imagination does help to generalize to novel sequences of operations as shown in the generalization gap of OCIM between the results of the *OOD op* split and the other baselines. (2) Current SOTA object-centric perception models like Slot Attention are not quite able to systematically generalize to objects composed of never-seen before arrangements of known arguments (such as new pairs of color/shape). This result is interesting and suggests that additional inductive biases or learning paradigms are needed to learn object-centric representations that disentangle independent factors of variations within an object. We did however notice that the choice architecture for the execution component seems to have an impact on the perception part, and that, surprisingly, both the GNN and GRU baselines perform better than OCIM on the OOD perception split.

**Modules Specialization.** We are also interested in analyzing how specialization of the learned modules (i.e. becoming activate for a certain operation) impacts the generalization performance of OCIM. For each training random seed, we count the number of times each module was selected for each of the ground-truth operations. Since we use the Gumbel softmax trick to select modules , we use the argmax of the attention coefficient to decide which module is selected. We report these proportions in the heatmap in Figure 4 for OCIM with and without imagination. The x-axis corresponds to the seed number; the y-axis corresponds to the module indices for both the addition and the subtraction operations.

The accuracies reported at the top of the heatmaps correspond to the validation and test accuracies on the *OOD op* splits (e.g. when evaluating the models on sequences of operations that have not been seen during training). We note the following three observations: (1) Current inductive biases are not sufficient for specialization to systematically occur since there are some seeds that have overlaps between selected modules. (2) As shown in the generalization results of OCIM-noim, specialization in the modules is not enough for the model to generalize to novel sequences of operations (e.g. modules in seed 2 of OCIM-noim (left column) are specialized yet we do not observe systematic generalization). Finally, (3) We notice that imagining new samples is necessary for the model to generalize, but also not sufficient as it needs the modules to be specialized as well. Seeds 0 and 5 of OCIM do have overlap in the module selections and the imagination framework was not successful. This observation explains the variance in terms of performance for the *OOD op* split that we report in Figure 3.

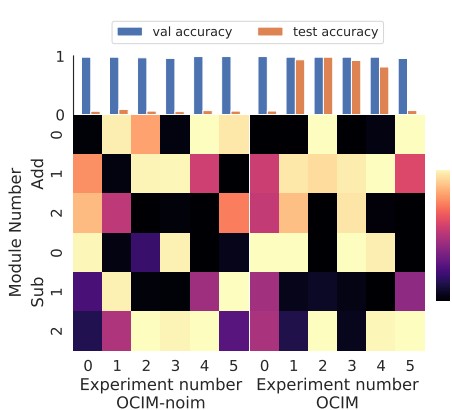

Figure 4: Modules specialization.

## 6 CONCLUSION AND FUTURE WORK

We have presented OCIM, a method that leverages object-centric representations to decompose visual reasoning tasks into a series of learned primitives (operation and object choices). OCIM combines these primitives in novel ways in order to generate and learn from unseen imaginary tasks, which radically improve OOD generalization. We compared OCIM against NPS and two other baselines without imagination on a synthetic visual arithmetic reasoning task in which we apply a sequence of operations to colored MNIST digits. We found that only OCIM was able to systematically generalize to new tasks composed of unseen sequences of arithmetical operations. Interestingly, we observed that specialization among the neural modules seems to be a necessary but not sufficient condition for modular architectures like OCIM and NPS to generalize to unseen sequences of operations: imagination seems to be a critical addition to the specialization condition. The effectiveness of imagination in our setup raises the question of whether its function is similar in biological brains. An interesting hypothesis is that dreams have a regularizing effect in the brain (OBH). While the link between OBH and OCIM is superficial, it poses an interesting question that might be worth exploring in future work.

Along with OCIM, we have introduced a synthetic visual reasoning benchmark to assess the extent to which imagination improves compositional generalization. Despite the simplicity of the benchmark, we found that SOTA models like NPS fail to compose the primitives learned during training in novel ways in order to generalize. As research in compositional generalization progresses, the benchmark could be extended with more challenging scenarios by increasing the number of operations, the length of the programs, and the number of objects.

**Reproducitbility Statement** We reported in the appendix all the model and training hyperparameters to implement and reproduce our model (Table 1, 2, 3, 4) as well as the detailed content of the different data splits. We will release the code and scripts for both our model, the baselines and the generation of the different dataset splits.

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

# A APPENDIX

## A.1 MODEL

**Controller.** The controller encodes a support set of I/O pairs $\mathcal{S} = \{\mathbf{X}, \mathbf{O}\}$ and outputs a task embedding $\mathbf{z}$ that is used to predict a task-specific output associated to a query input. We formulate the controller in an iterative manner: it starts with a random guess $\mathbf{z}_{\text{init}}$ (sampled from a learned gaussian) and refines it. We denote Refine$(\mathbf{z}, \mathcal{S})$ a refinement step and $T$ the total number of refinement steps. A refinement step at timestep $t$ can be decomposed into three main steps:

- **Step 1** - Current guess prediction: we compute the current predicted outputs $\hat{\mathcal{O}}$ associated with each input in the support set s.t. $\hat{\mathcal{O}}_t = \{\text{ComputeGuess}(\mathbf{X}, \mathbf{z}_t)\}$
- **Step 2** - Compare current guess $\hat{\mathcal{O}}_t$ to ground-truth outputs $\mathbf{O}$ given their associated inputs in $\mathcal{S}$. We denote $\mathbf{a}_t = \text{Compare}(\hat{\mathcal{O}}_t, \mathcal{S}, \mathbf{z}_t)$ the output of this step.
- **Step 3** - Update current task embedding $\mathbf{z}_{t+1} = \text{Update}(\mathbf{a}_t, \mathbf{z}_t)$

ComputeGuess$(\mathbf{X}, \mathbf{z}_t)$ is model-specific and simply consists of predicting the current task output given $\mathbf{z}_t$. And $\mathbf{z}_{t+1} = \text{Update}(\mathbf{a}_t, \mathbf{z}_t)$ is parametrized as a simple GRU RNN taking $\mathbf{a}_t$ as input and updating the hidden state $\mathbf{z}_t$. We then need to detail how we obtain $\mathbf{a}_t = \text{Compare}(\hat{\mathcal{O}}_t, \mathcal{S}, \mathbf{z}_t)$ This is done in two steps (1) - first we compute a representation $\mathbf{b}_i^t$ for each sample $i$ in the support set, then, (2) we aggregate those representations to obtain $\mathbf{a}_t$. To do so, each input $\mathbf{x}_i \in \mathbf{X}$ is transformed into an object-centric set of slots $\{\mathbf{s}_j^i\}_j$ using the same perception model as in the rest of the model. Each slot is then concatenated to the current guess $\mathbf{z}_t$, the ground-truth associated input $\mathbf{o}_i$ and the currently predicted output $\hat{\mathbf{o}}_i$. The sample-wise result is then obtained with a simple GRU RNN. We denote this step by $\mathbf{b}_i^t = \text{EncodeSample}(\mathbf{x}_i, \mathbf{z}_t, \mathbf{o}_i, \hat{\mathbf{o}}_i)$ and the resulting sequence representation by $\mathbf{b}_t = [\mathbf{b}_i^t]$.

We then need to aggregate the obtained results accross the whole support set. To do so, we concatenate each sample representation $\mathbf{b}_i^t$ with the current task embedding $\mathbf{z}_t$, the ground-truth associated input $\mathbf{o}_i$ and the currently predicted output $\hat{\mathbf{o}}_i$. Similarly we aggregate the results with a simple GRU RNN. We denote $\mathbf{a}_t = \text{EncodeSupport}(\mathbf{b}_t, \mathbf{z}_t, \mathbf{O}, \hat{\mathbf{O}}_t)$. The exact parametrization of each of the modules consituting the controller is given in Table 1.

| Description | Symbol | Architecture |
|---|---|---|
| Task embedding refiner | $\text{Update}(\mathbf{a}_t, \mathbf{z}_t)$ | GRUCell(64, 64) |
| Sample concatenation before aggregation | $h_j^t(i). = \text{Concat}(\mathbf{s}_j^i, \mathbf{z}_t, \mathbf{o}_i, \hat{\mathbf{o}}_{it})$ | MLP(128 + 10, 64) |
| Sample representation aggregation accross slots | $b_i^t = Agg([\hat{h}_j^t(i)])$ | GRU(64, 64) |
| Sample-wise processing before aggregation | | MLP(64, 64) |
| Aggregation accross support elements | $\mathbf{a}_t = \text{EncodeSupport}(\mathbf{b}_t, \mathbf{z}_t, \mathbf{O}, \hat{\mathbf{O}}_t)$ | GRU(64, 64) |

Table 1: Controller Hyperparameters

**Perception.** The perception component is implemented with a Slot Attention module. We keep the same hyperparameters as the initial version (Locatello et al., 2020) with a slot-wise dimension of $64$ and using different Gaussian parameters for each slot. We perform all the experiments with $4$ slots.

**Selection Bottleneck.** The selection bottleneck is composed of three parts: (1) the module that transforms a single task embedding $\mathbf{z}$ into a sequence of embeddings from which the step-wise elements of the neural program will be predicted, (2) the extraction of keys, queries, and values for the condition and the module selection, and (3) the step-wise gate prediction to determine the number of updates to perform (e.g. neural program length). We detail the parameterizations of these three parts in Table 2

| Description | Symbol/Notation | Architecture/Value |
|---|---|---|
| Sequence prediction | $[\mathbf{z}_1, ...\mathbf{z}_T] = \text{Pred}(\mathbf{z})$ | GRU(64, 64) |
| Gate prediction | $g_t = \text{Gate}(\mathbf{z}_t)$ | $\sigma(\text{MLP}(64, 1))$ |
| Query prediction for module/condition selection | $Q_r(\mathbf{z}_t)$ and $Q_c(\mathbf{z}_t)$ | MLP(64, 64), MLP(64, 64) |
| Learned Module/Condition Key embedings | $\mathbf{M}$ and $\mathbf{C}$ | Embeding($N_r$, 64) and Embeding($N_c$, 64) |
| Max number of steps | $T$ | 2 |
| Gumbel softmax temperature | temp | 3 |

Table 2: Selection Bottleneck Hyperparameters

**Executor.** The executor is composed of three main parts: (1) the learned neural program library (e.g. modules and condition vectors), (2) the argument selection part given a condition vector, and the (3) task-related output prediction. We detail the parameterization of these part in Table 3.

| Description | Symbol/Notation | Architecture/Value |
|---|---|---|
| Modules | $[\mathbf{m}_1, ...\mathbf{m}_{N_r}]$ | [GRU(64, 64)] |
| Condition vectors | $[\mathbf{c}_1, ...\mathbf{c}_{N_c}]$ | Embedding($N_c$, 64) |
| Condition/Modules keys | $\mathbf{C}$ and $\mathbf{M}$ | Embedding($N_c$, 64) and Embedding($N_r$, 64) |
| Key prediction for argument selection | $K_a(\mathbf{S})$ | MLP(64, 64) |
| Output prediction | $y = \text{TaskPred}(\mathbf{h}_T)$ | $\sigma(\text{MLP}(64, \text{num bits}))$ |
| Number of conditions/modules | $N_c$ and $N_r$ | 3 and 3 |

Table 3: Execution component Hyperparameters

## A.2 TRAINING

The training of the whole model can be split into three phases:

- **Step 1**: Pretraining of the perception model such that the next steps start with reasonable latent slots.

- **Step 2**: Regular training on the task prediction objective (here 8-binary-bits digit prediction).

- **Step 3**: Imagination, where random modules and conditions are sampled to create new data points and expose the model to potentially OOD samples.

Each of these steps gives rise to a specific objective loss to optimize, namely: $\mathcal{L}^{\text{rec}}$, $\mathcal{L}^{\text{task}}$, and $\mathcal{L}^{\text{im}}$. The different phases consist of adding progressively these losses to the optimized objective. The total loss is:

$$\mathcal{L} = \mathcal{L}^{\text{task}} + \alpha\mathcal{L}^{\text{rec}} + \beta\mathcal{L}^{\text{im}}$$

The pretraining of the slot attention modules (Step 1) is done separately and we initialize it with the pretrained weights when adding the task-specific loss at Step 2. The coefficient $\alpha$ is fixed, whereas $\beta$ is introduced after a warm-up period and increased linearly during a certain number of epochs. We report these training-specific hyperparameters in Table 4

| Description | Symbol/Notation | Value |
|---|---|---|
| Reconstruction coefficient | $\alpha$ | 0.0002 |
| Imagination coefficient | $\beta$ | 50 |
| Warmup period before imagination | | 450 epochs |
| Imagination coef schedule | | from 0 to $\beta$ in 200 epochs |
| Batch size | | 32 |
| Learning rate | $l_r$ | $2e^{-4}$ |

Table 4: Training Hyperparameters

**Dataset** For all the different splits, we trained OCIM on 5000 samples where each sample had 10 I/O pairs in the support set and 1 I/O query example. Validation and Test set are composed of 100 and 1000 samples respectively. We describe below the generating steps of the three different splits we considered in the experiments:

- *OOD op*: models are trained on sequences of operations $\{[+], [-], [+-], [++], [--]\}$ and evaluated on $[-+]$. The model is also trained and evaluated on one order only: top left, top right, then bottom. Digits, signs and colors are sampled randomly iid.

- *OOD order*: models are trained on all sequences of operations $\{[+], [-], [+-], [++], [--], [-+]\}$ and on one object order (top left, top right, then bottom); they are then evaluated on all operations, and all possible orders in which the visual digits can be considered.

- *OOD percep*: for each digit, only two (sampled in a uniform way) possible positive colors and two negative colors are considered during training. The models are evaluated on the left-out color configurations. Models are trained and evaluated on all sequences of operations$\{[+], [-], [+-], [++], [--], [-+]\}$

