# OpenReview forum: "OCIM : Object-centric Compositional Imagination for Visual Abstract Reasoning"
_ICLR.cc/2023/Conference — Submitted to ICLR 2023_

### Official Review · Reviewer_Czgi · 2022-10-24

**Confidence:** 4
**Correctness:** 2
**Technical Novelty And Significance:** 3
**Empirical Novelty And Significance:** 2
**Recommendation:** 3

**Clarity, Quality, Novelty And Reproducibility:**

The main method part (Sec 4) is hard to follow. I would suggest having a running example that helps to explain what each step is doing. Also, the captions of the figures need to be improved by including more explanations to help readers understand better.

Besides, I could not find concrete numerical values of the results in Figures 3 and 4, not even in the appendix. Lacking concrete numbers makes readers hard to judge the extent of the improvement and reproduce the results. It would also benefit the reproducibility if the training curves could be provided (maybe in the appendix).

The idea of object-centric imagination is novel, but not well-supported by experiments.

**Strength And Weaknesses:**

Pros:
1. This work proposes a new synthetic visual reasoning benchmark that provides access to operations, operation order, objects primitives.
2. The novel OCIM model presents better generalization ability over other baselines on the new benchmark.

Cons:
1. The writing needs to be improved, detailed in the clarity part.
2. Lacking ablation studies make it hard to estimate the importance and functionality of each module. May consider the following (you may put details in the appendix and state the ablation study results briefly in the main text if space is not allowed):
* What if $\beta$ (imagination coefficient) is set to zero (disabling the imagination)?
* How important are the warmup epochs?
* Can you modify the max number of steps $T$? $T=2$ seems a very specific design for the benchmark task.
3. The evaluations are only done on a self-proposed dataset that provides access to primitives. It is unclear how this method can be generalized to other more challenging benchmarks or benchmarks that do not have access to these primitives.
4. The overall motivation is clearly stated. However, the motivations behind the specific design of the OCIM model need more explanation.

**Summary Of The Paper:**

This work proposes Object-centric Compositional Imagination (OCIM) for visual reasoning tasks. It decomposes the tasks into a series of primitives applied to objects and recomposes these primitives to imaginary tasks. The model learns to reuse the previously-learned concepts to systematically generalize at test time by training on these imagined tasks. The model shows better generalization ability than the other baselines on a synthetic visual arithmetic reasoning benchmark the authors newly proposed.

**Summary Of The Review:**

It is an interesting and novel idea to utilize the object-centric imagination method to improve the OOD generalization of visual reasoning tasks. However, the paper's writing and supporting experiments need to be improved with more effort.

---
Update (2022/11/22): I would keep my score before new results provide.

---

> ### Author Response · Authors · 2022-11-18
> **Ablations and model computational steps clarity**
>
> We thank the reviewe for the useful comments and adressed some of the weaknesses pointed out below :
>
> -  **Lacking ablation studies make it hard to estimate the importance and functionality of each module. May consider th e following (you may put details in the appendix and state the ablation study results briefly in the main text if space is not allowed):[..]**
> Beta coefficient set to zero correspond to NPS-adapt baseline (we updated the text in section 5.1 and called it OCIM-noim instead to avoid any confusions).
> We’re running the other suggested ablations (warmup epochs, number of max steps) along with an ablation of the number of modules/ conditions.  Concerning the number of warmup epochs we found that the model needed a minimum number of them (to learn meaningful modules/conditions that can be later composes) but once a reasonable accuracy is reached on the training data, the model was not sensitive to increasing the number of warmup epochs.
>
> - **The evaluations are only done on a self-proposed dataset that provides access to primitives. It is unclear how this method can be generalized to other more challenging benchmarks or benchmarks that do not have access to these primitives.**
> Note that, even with access to the primitives (though hidden from the model), the task is very hard and current methods are far from achieving optimal generalization performance. In addition, we believe that the insights derived from our work could inspire future work on more challenging scenarios.
>
> - **The overall motivation is clearly stated. However, the motivations behind the specific design of the OCIM model need more explanation.
> The main method part (Sec 4) is hard to follow. I would suggest having a running example that helps to explain what each step is doing. Also, the captions of the figures need to be improved by including more explanations to help readers understand better.**
> Thank you, we have updated the beginning of Section 4  with the computational steps of OCIM for more clarity.
>
> - **Besides, I could not find concrete numerical values of the results in Figures 3 and 4, not even in the appendix. Lacking concrete numbers makes readers hard to judge the extent of the improvement and reproduce the results. It would also benefit the reproducibility if the training curves could be provided (maybe in the appendix).**
> We will add those to the appendix once the ablations are finished.

---

### Official Review · Reviewer_Zv6d · 2022-10-25

**Confidence:** 4
**Clarity, Quality, Novelty And Reproducibility:** See my weakness comment #1 for my sug…
**Correctness:** 3
**Technical Novelty And Significance:** 3
**Empirical Novelty And Significance:** 2
**Recommendation:** 5

**Strength And Weaknesses:**

The paper presents an approach for neural-program-based approach for solving ARC tasks. The proposed approach is well-motivated, and shows improvements in experiments. My main concerns about the paper are the following three.

First, I think the paper writing can be significantly improved. The major issues are:
1. Figure 2 is not an overview of the model. For example, it does not contain the object perception module (which the authors claim is an object they want to study in their contribution #3)
2. Section 4 does not contain a full description of the proposed method. I have to look at Algo 3 and based on the names to guess what's used for object perception.
3. The current writing of the paper is very unclear about the contribution of the authors. For example, my understanding that 4.1 and 4.2 are not the contribution of the author (See section 5.1: Our model OCIM can be seen as an implementation of the sparse interaction inductive biases proposed in Neural Production Systems (NPS) Goyal et al. (2021a), but augmented with an imagination-based learning mechanism.)

Second, given that the neural programming model is not the contribution of the paper, my understanding is that the paper focuses on "imagination." While overall I very much like the idea of using trained models to generate new data, and the authors have found an interesting domain to execute the idea, I am a bit worried about the generality of the proposed method.

Specifically, it seems that the dataset is constrained to a "sequential program" structure. And furthermore, almost all possible combinations of learned "modules" can be executed on this dataset (e.g., different orders, different visual positions). However, this is generally not true for many domains. There are at least two cases.

First, consider we want to use a program to query "the color of digit 1." The program assumes the existence of a digit 1 in the data, which, if not satisfied, will lead to unexpected execution results. I can easily imagine that adding those data into the training set will hurt model performance. I think the authors should at least talk about these assumptions about their domain, and discuss why their methods can work in the specific domain. That being said, I think the proposed method is too constrained.

Second, when these modules have dependencies among each other, it is unclear how we can sample programs that are "valid."

Third, the experimental results of this paper are a bit weak, due to the limitation to one dataset curated by the authors themself. Due to the limitations, I described above, I think it is important to extend the results to other domains, for example, other ARC tasks, or tasks such as visual question answering etc.

Other than these limitations, I think the authors should also cite more related works.

For example, the authors have clearly missed several important works in neural programming, such as
- Neural Programmer-Interpreters, https://arxiv.org/abs/1511.06279
- Making Neural Programming Architectures Generalize via Recursion https://arxiv.org/abs/1704.06611
- Closed Loop Neural-Symbolic Learning via Integrating Neural Perception, Grammar Parsing, and Symbolic Reasoning http://proceedings.mlr.press/v119/li20f/li20f.pdf

The contribution of "disentanglement of object properties in object-centric representation learning" is closely related to many visual representation work. Here are some pointers:
- https://cs.stanford.edu/people/jcjohns/clevr/ Many works have worked on the CLEVR-CoGenT split, which is exactly the setup you have been studying.
- ImageNet-trained CNNs are biased towards texture; increasing shape bias improves accuracy and robustness. https://arxiv.org/abs/1811.12231

**Summary Of The Paper:**

This paper presents a neural program-based method for an abstract reasoning (ARC from Chollet 2019) style task. The model contains an object-centric representation layer (based on slot attention I believe), a controller (encoding the examples and predicting the program), and an executor (which takes the program representation as input and predicts the final answer). The claimed contributions are two-folds. First, the program representation (which is a sequence of module indices, and a sequence of control parameters). Second, the training paradigm, which includes a "dream" stage.

**Summary Of The Review:**

I like the idea of the paper on dreaming with programmatically structured neural networks for generalization. However, the results of the paper are limited. The paper writing can still be improved. Overall my recommendation is WR.

---

> ### Author Response · Authors · 2022-11-18
> **Writing updated**
>
> We thank the reviewer for the useful comments. We addressed the weaknesses pointed out below :
>
> - **Figure 2 is not an overview of the model. For example, it does not contain the object perception module (which the authors claim is an object they want to study in their contribution #3)**
> Our method is agnostic to the object perception module. For completeness, we have updated Figure 2 and added the Slot Attention perception module.
>
> - **Section 4 does not contain a full description of the proposed method. I have to look at Algo 3 and based on the names to guess what's used for object perception.**
> We updated the first paragraph of the model in Section 4 to include a full overview of the computational steps of our model.
>
> - **The current writing of the paper is very unclear about the contribution of the authors. For example, my understanding that 4.1 and 4.2 are not the contribution of the author (See section 5.1: Our model OCIM can be seen as an implementation of the sparse interaction inductive biases proposed in Neural Production Systems (NPS) Goyal et al. (2021a), but augmented with an imagination-based learning mechanism.)**
> Thank you for pointing this out, sections 4.1 and 4.2 contain some differences with respect to NPS that were not explicitly introduced as a contribution in our work.
> NPS was taken as a reference for the sparse interaction inductive biases it proposes : NPS sequentially updates a set of slots by choosing at each time-step a primary slot, a secondary slot, and an interaction module with some key-query attention mechanisms. Whereas OCIM sequentiay updates a working memory (from which the answer will be retrieved)  by selecting among the input set of slots an argument and an update module with some key-query attention mechanisms.
> An important distinction is how the key-query selection mechanism is done. To be able to hallucinate new “programs” that are independent from the input sample, we propose using a controller and an executor. The controller generates programs descriptions and the selection bottleneck (4.1)  translates it into sequential programs that the executor (4.2) follows to generate an answer. Note that in NPS this happens implicitly since the input is directly used to attend the rules that are applied on itself with a self-attention mechanism. Thus, with NPS it is not possible to execute the rules inferred for one sample to another one, which prevents it from being able to use imagination mechanisms. We have updated section 5.1 to reflect this difference and highlight our contribution.
>
> - **Second, given that the neural programming model is not the contribution of the paper, my understanding is that the paper focuses on "imagination." While overall I very much like the idea of using trained models to generate new data, and the authors have found an interesting domain to execute the idea, I am a bit worried about the generality of the proposed method.**
> In order to address the concern about the generality of the proposed we will increase the diversity of tasks and query that can be made and composed in our benchmark to make it more general.
>
> - **Specifically, it seems that the dataset is constrained to a "sequential program" structure. And furthermore, almost all possible combinations of learned "modules" can be executed on this dataset (e.g., different orders, different visual positions). However, this is generally not true for many domains. There are at least two cases.**
> Thanks for pointing this out. Our method proposes a sequential form of update which we think is general enough since a vast majority of programs can be translated into a sequential process to execute and produce an answer.
> - **First, consider we want to use a program to query "the color of digit 1." The program assumes the existence of a digit 1 in the data, which, if not satisfied, will lead to unexpected execution results. I can easily imagine that adding those data into the training set will hurt model performance. I think the authors should at least talk about these assumptions about their domain, and discuss why their methods can work in the specific domain. That being said, I think the proposed method is too constrained.**
> This example can be addressed by adding a “default” or “None” answer. We will consider adding those cases to the extended version of the dataset.
>
> - **Second, when these modules have dependencies among each other, it is unclear how we can sample programs that are "valid."**
> This is true. This can be solved by adding a prior over modules/conditions that can be learned during training (KL term) and sampling from it instead of a uniform prior.

---

> > ### Author Response · Authors · 2022-11-18
> > **Part 2**
> >
> > - **Third, the experimental results of this paper are a bit weak, due to the limitation to one dataset curated by the authors[..] I think it is important to extend the results to other domains, for example, other ARC tasks, or tasks such as visual question answering etc.**
> > Other Visual abstract reasoning domains have a discriminative task. We wanted to put the focus on a generative benchmark. ARC would indeed be a strong addition but it has the added difficulty of images generation as an answer that can be of any size.
> > The main motivation for proposing this benchmark was to design a generative benchmark (where the answer has to be generated and not selected among a set of candidate answers) that would be more approachable by deep learning methods than the ARC dataset. [1] also advocates, in a recent review on abstraction and analogy in artificial intelligence, for generative abstract reasoning datasets and argues that models trained to solve them are less prone to shortcut learnings and are generally more interpretable. We updated the dataset motivation section 3. to that end. We will however diversify the set of tasks and primitives that can be composed.
> >
> > [1]Melanie Mitchell - Abstraction and Analogy making in Artificial Intelligence - https://arxiv.org/pdf/2102.10717.pdf

---

> ### Comment · Reviewer_Zv6d · 2022-11-24
> **Response**
>
> Thank you for the response. I think the revisions to the paper have greatly improved the clarity of the technical contributions. However, as I discussed, my primary concern is that there are still many potential issues with the proposed method when considering its generalization to more practical domains.
>
> The authors have proposed several ideas for resolving these challenges, but I don't think they can be easily addressed. For example:
>
> - "This example can be addressed by adding a “default” or “None” answer. We will consider adding those cases to the extended version of the dataset." How would you get the training data for such questions? Does this mean you need to have annotations for questions that are "unanswerable?" This would be a much stronger assumption compared to other works.
>
> - "This is true. This can be solved by adding a prior over modules/conditions that can be learned during training (KL term) and sampling from it instead of a uniform prior." I am not sure how this can be learned. Specifically, my understanding is that these modules are learned and "latent," so I don't understand how authors can learn that two modules **cannot** be combined: that is, when they get combined during "imagination," they will produce, e.g., "meaningless" outputs.
>
> Given the limited contribution of this paper over existing work (NPS) and the fact that the main contribution of the paper is about imagination-based training, I think these issues should be addressed before this paper gets published.

---

### Official Review · Reviewer_65Tv · 2022-10-25

**Confidence:** 4
**Correctness:** 2
**Technical Novelty And Significance:** 2
**Empirical Novelty And Significance:** 2
**Recommendation:** 3

**Clarity, Quality, Novelty And Reproducibility:**

The proposed method is novel, but the experimental evaluation is insufficient to support the claims.

**Strength And Weaknesses:**

## Strengths
The proposed imagination-based learning is interesting and able to improve the systematic generalization over a certain apsect (i.e., sequence of operations)

## Weaknesses

1. It is unclear how to randomly sample a new program, i.e., from what probability distributions the gates, the condition vectors, and the processing modules are sampled.

2. The authors claim the proposed imagination method as "Object-centric Compositional Imagination". However, it seems that the proposed imagination methond (described in Section 4.3) can also be used to non-object-centric representations, e.g., grid representations of images. The authors do not conduct any experiment using non-object-centric features.

3. The proposed visual arithmetic reasoning dataset is similar to RAVEN-style datasets [1], i.e., few-shot learning a rule from a support set and applying it to a new input. The RAVEN-style datasets have been extensively studied by previous works. It is unclear why it is necessary to synthesize such a new dataset. Besides, the authors only conduct experiments on the proposed dataset and do not compare with related works in RAVEN.

[1] Zhang, Chi, et al. "Raven: A dataset for relational and analogical visual reasoning." Proceedings of the IEEE/CVF Conference on Computer Vision and Pattern Recognition. 2019.

4. Figure 3 is the only experimental result presented in this paper and from this figure, we can see that the proposed method has a much larger variance than baselines. There is not any explanation about this phenomenon in the paper.


**Summary Of The Paper:**

This paper focuses on the compositional generalization of visual abstract reasoning, i.e., the  ability  to  compose learned  concepts  in  novel  ways  for new situations. The authors propose an imagination-based learning framework applied on object-centric representations. By training on the imagined tasks, the model is expected to obtain better systematic generalization at test time.

To evaluate the proposed framework, the authors synthesize a new visual arithmetic reasoning benchmark. Specifically, each example in the benchmark contains a support set of 10 input-output pairs and the outputs of these 10 pairs are obtained by applying a same latent program to the corresponding inputs.  Based on the given support set, the model is required to predict the output of a query input. The inputs are images with three colored MNIST digits placed at three different positions and the colors represent the signs of the digits. The program applied to the inputs is a sequence of arithmetic operations (i.e., addition and subtraction) in a particular positional order. There are three splits in the benchmark to evaluate generalization to new sequences of operations, position orders, and digit-Color configurations.

The used model to solve the proposed benchmark consists of a controller, which generates a latent task embedding based on the given support set, and an executor, which generates a neural program based on the task embeding and then apply the generated neural program to predict the output of the query input. The proposed imagination learning works by (1) randomly sampling neural programs, (2) applying the sampled neural programs to the inputs of original samples to generate new samples, and (3) training the model on the new samples. Experiments show that the imagination learning can improve the generalization to new sequences of operations.







**Summary Of The Review:**

Although the proposed method is novel, the experimental evaluation is insufficient. Therefore, I recommend rejection.

-------post-rebuttal-----
Thank the authors for the response. The response solves part of my concerns. Still, the main concern about the insufficient experiments is not resolved. Therefore, I will keep my rating. Hope the authors could further improve their work in the future.

---

> ### Author Response · Authors · 2022-11-18
> **Clarifications on the object-centric side of the model, variance and comparison with other visual reasoning benchmarks**
>
> We thank the reviewer for those useful comments and addressed some of the weaknesses pointed out below :
>
> - **It is unclear how to randomly sample a new program, i.e., from what probability distributions the gates, the condition vectors, and the processing modules are sampled.**
> Thanks for pointing this out, We added that clarification in the text : each condition, module and gate sequences are sampled from a uniform distribution.
> - **The authors claim the proposed imagination method as "Object-centric Compositional Imagination". However, it seems that the proposed imagination methond (described in Section 4.3) can also be used to non-object-centric representations, e.g., grid representations of images. The authors do not conduct any experiment using non-object-centric features.**
> In general, any discrete decomposition of an image could be loosely considered as object-centric. Thus, even a grid representation of an image would be object-centric (eg. where objects are pixels).
> However for our method to work (ie. for the model to learn “meaningful” modules that take “meaningful” objects as argument for the task at hand), it needs to operate on object-centric representations where the input arguments that the modules can select correspond to objects that make reasoning easy for the task at hand.
> If the input argument are pixel positions, the model will need to learn many more modules that would correspond to pixel-level transformation and composing them during imagination would not result in useful imagined programs for the task at hand.
> - **The proposed visual arithmetic reasoning dataset is similar to RAVEN-style datasets [1], i.e., few-shot learning a rule from a support set and applying it to a new input. The RAVEN-style datasets have been extensively studied by previous works. It is unclear why it is necessary to synthesize such a new dataset. Besides, the authors only conduct experiments on the proposed dataset and do not compare with related works in RAVEN.**
> The main motivation for proposing this benchmark was to design a generative benchmark (where the answer has to be generated and not selected among a set of candidate answers) that would be more approachable by deep learning methods than the ARC dataset. [1] also advocates, in a recent review on abstraction and analogy in artificial intelligence, for generative abstract reasoning datasets and argues that models trained to solve them are less prone to shortcut learnings and are generally more interpretable. We updated the dataset motivation section 3. to that end.
> We will however diversify the set of tasks and primitives that can be composed.
>
> - **Figure 3 is the only experimental result presented in this paper and from this figure, we can see that the proposed method has a much larger variance than baselines. There is not any explanation about this phenomenon in the paper.**
> Thank you for pointing this out. The proposed method combines multiple recurrent modules and thus it is sensitive to gradient vanishing and explosion problems. This makes it highly sensitive to initialization. Despite this, OCIM is clearly a promising approach for solving problems with OOD operations as seen in Figure 3.
> In Figure 4 we give an explanation for this variance as we notice that seeds that do not generalize well in OCIM have learned modules that are not specialized. We however show in this same figure that specialization is necessary but not sufficient since if we remove the imagination regime, seeds that result in specialized modules are not able to generalize. The model needs the imagination objective to be trained to  “recognize” OOD samples. We updated the caption of Figure 3 to in that regard to reference the paragraph in section 5.3 that give some insights about the high variance of the generalization results.
>
> [1]Melanie Mitchell - Abstraction and Analogy making in Artificial Intelligence - https://arxiv.org/pdf/2102.10717.pdf

---

### Official Review · Reviewer_zq7u · 2022-10-26

**Confidence:** 3
**Correctness:** 2
**Technical Novelty And Significance:** 2
**Empirical Novelty And Significance:** 2
**Recommendation:** 3

**Clarity, Quality, Novelty And Reproducibility:**

The paper is generally well written, however, the notation in Eq.3 and 4 should probably be revised to include T-1 to avoid confusion about the current and prior steps in RNN/GRU.



**Strength And Weaknesses:**

Strengths:
- The proposal of the imagination framework is interesting working within the latent space of the model, which should provide increased generalisation without being limited to generation.
- The approach is fully differentiable that is and advantage against some prior methods for exutors.
- The dataset can be useful if well-defined against prior datasets and models
Weaknesses:
- The augmentation strategy can be broadly compatible with data augmentation strategies which don't seem to have been considered in related work or to inspire the approach.
- The data augmentation is loosely out of distribution, as the numerical dataset has a relatively limited scope with a simple range of numbers and operations. As the explicit function of the functions is unknown, and the logical operations can be deduced as a combinational operation that is unclear if possible in the leave-out strategy.
- It would have been interesting for the approach to be applied to standard Abstract reasoning datasets such as RAVEN. While they are different problems, they could have shown greater generalisation and provided insights into how the model is working.
- Similar to the above, the dataset seems highly similar to RAVEN and others, just numbers instead of shapes. It isn't clear where this dataset contributes.

**Summary Of The Paper:**

The paper proposes a method for imagination (data augmentation) to improve the generalisation of abstract reasoning approaches. The general module is based on adapting slot-attention to work over a sequence of to select the latent function to apply. The augmentation works using Gumbel soft max trick to select the augmentation. The authors additionally propose a new dataset of mathematical problems with values, colours and operations to be resolved. They compare to the prior approach on NPS overfitting to training but improving generalisation on test.

**Summary Of The Review:**

While the paper works on a challenging problem and shows generalisation across unseen problems based on their proposed baselines. The lack of rigorous evaluation and benchmarks from similar problems means the paper feels incomplete. Additional experiments or justification for their lack of inclusion needs to be provided to position this paper.

---

> ### Author Response · Authors · 2022-11-18
> **On the comparison with other visual reasoning benchmarks**
>
> We thank the reviewer for the useful comments, we addressed some of the weaknesses pointed out below : .
>
> - **The augmentation strategy can be broadly compatible with data augmentation strategies which don't seem to have been considered in related work or to inspire the approach.**
> Thank you for the suggestion. Could you point us to some works that we could reference in our work?
>
> - **The data augmentation is loosely out of distribution, as the numerical dataset has a relatively limited scope with a simple range of numbers and operations. As the explicit function of the functions is unknown, and the logical operations can be deduced as a combinational operation that is unclear if possible in the leave-out strategy.**
> We will add more complicated primitives than can be composed with the operations that are currently present in the dataset. Could you be more specific with what you mean by "the leave-out strategy?"
>
> - **It would have been interesting for the approach to be applied to standard Abstract reasoning datasets such as RAVEN. While they are different problems, they could have shown greater generalisation and provided insights into how the model is working. Similar to the above, the dataset seems highly similar to RAVEN and others, just numbers instead of shapes. It isn't clear where this dataset contributes.**
>  The main motivation for proposing this benchmark was to design a generative benchmark (where the answer has to be generated and  not selected among a set of candidate answers) that would be more approachable by deep learning methods than the ARC dataset.
> [1] also advocates, in a recent review on abstraction and analogy in artificial intelligence,  for generative abstract reasoning datasets and argues that models trained to solve them are less prone to shortcut learnings and are generally more interpretable.
> We updated the dataset motivation section 3. to that end.
>
>
> [1] Melanie Mitchell - Abstraction and Analogy making in Artificial Intelligence - https://arxiv.org/pdf/2102.10717.pdf

---

### Decision · Program_Chairs · 2023-01-20

**Decision:**

Reject

**Justification For Why Not Higher Score:**

Unanimous agreement that the paper has not performed sufficiently rigorous evaluation, an issue acknowledged by the authors themselves.

**Justification For Why Not Lower Score:**

n/a

**Metareview: Summary, Strengths And Weaknesses:**

This paper proposes Object-centric Compositional Imagination (OCIM) to improve the generalization of visual abstract reasoning. Specifically, it's about "imagination" and the ability to compose concepts to address novel situations. By training on "imagined" data, the premise is that better generalization could be seen at test time. The neural program-like method has an object-centric representation layer, a controller and an executor. It decomposes the tasks into a series of primitives, then composes these to new imaginary tasks. Specifically, it randomly samples neural programs, then applies the programs to the original data in order to create new data, and then train on the new data. In terms of results, performance of the model is better than baselines on a new synthetic benchmark proposed in this paper also.


-- STRENGTHS --

The proposed framework is novel and interesting, and is able to improve generalization (compared to other baselines) on unseen problems, to a certain extent.


-- WEAKNESSES --

There is unanimous agreement that the paper fares poorly in terms of rigorous evaluation. Specific pieces of evidence to support this are:

1) Evaluation is performed only on a dataset created by the authors themselves, without any evaluation on related datasets or tasks (e.g. RAVEN, ARC, etc.)

2) Limited amount of detailed evaluation, such as lack of ablation studies, makes it difficult to understand the characteristics and limitations of the model overall, as well as the contributions of various design decisions.

3) The author-proposed dataset provides access to primitives, which raises questions about whether the proposed method can generalize well to other datasets that do not do so.

The authors have acknowledged these concerns and have not rebutted per se. Their responses have been along the lines of "in order to address the concern about the generality of the proposed we will increase the diversity of tasks and query that can be made and composed in our benchmark to make it more general."

In light of these, it is clear that the paper should not be accepted in its current form.


**Summary Of Ac-Reviewer Meeting:**

n/a